# In Silico Analysis of the MitraClip in a Realistic Human Left Heart Model

Salvatore Pasta [1,2] 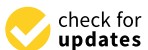

1   Department of Engineering, Università degli Studi di Palermo, Viale delle Scienze Ed.8, 90128 Palermo, Italy; salvatore.pasta@unipa.it; Tel.: +39-09123897277
2   Department of Research, IRCCS-ISMETT, 90100 Palermo, Italy

**Abstract:** Mitral valve regurgitation is a common heart valve disorder associated with significant morbidity and mortality. Transcatheter mitral valve repair using the MitraClip device has emerged as a safe and effective alternative for patients unsuitable for conventional surgery. However, the structural and hemodynamic implications of MitraClip implantation in the left ventricle have not been extensively explored. This study aimed to assess the structural and hemodynamic performance of the MitraClip device using a high-fidelity model of the human heart, specifically focusing on a healthy mitral valve geometry. The implantation of the MitraClip device was simulated using the finite element method for structural analysis and the lattice Boltzmann method for computational flow analysis. MitraClip implantation induced geometrical changes in the mitral valve, resulting in local maxima of principal stress in the valve leaflet regions constrained by the device. Hemodynamic assessment revealed slow-moving nested helical flow near the left ventricular wall and high flow velocities in the apex regions. Vorticity analysis indicated abnormal hemodynamic conditions induced by the double-orifice area configuration of the mitral valve after MitraClip implantation. By predicting possible adverse events and complications in a patient-specific manner, computational modeling supports evidence-based decision making and enhances the overall effectiveness and safety of transcatheter mitral valve repairs.

**Keywords:** MitraClip; cardiac mechanics; finite element analysis; computational fluid dynamic



## 1. Introduction

Mitral valve regurgitation is a common heart valve disorder with significant morbidity and mortality [1]. Its incidence is estimated at two to three cases per 1000 person years, affecting approximately 1% to 2% of the general population [2]. Improved life expectancy is expected to increase in prevalence. Management depends on the severity and type of regurgitation. Pharmacological therapy may be inadequate for degenerative mitral valve cases, making cardiac surgery the primary effective option. Conversely, functional regurgitation is usually treated according to the underlying cardiomyopathy.

The advent of transcatheter mitral valve repair offers a safe and effective, minimally invasive solution for treating diseased mitral valves [3]. This percutaneous approach is preferred for patients ineligible for conventional surgery due to age or contraindications. Transcatheter mitral valve therapies are based on annuloplasty and the MitraClip, which enables edge-to-edge repair of the valve leaflets under transesophageal echocardiographic guidance [4]. The latest generation of MitraClip systems has shown impressive results, reducing mitral regurgitation severity to less than two in 96.6% of patients within 20 days post implantation [5]. The device's capability to independently grasp the anterior and posterior leaflets has contributed to higher success rates, especially in challenging anatomies. Variations in clinical trial outcomes of the MitraClip are attributed to pathological conditions, operator experience, mitral valve anatomy, and patient selection, making pre-operative planning particularly challenging [6]. Patient-specific modeling of structural

heart interventions has proven valuable in understanding the biomechanical interactions between the device and the human host, especially in transcatheter aortic valve implantation settings [7,8] and diseased mitral valve conditions [9–11]. Simulations of the MitraClip demonstrate the device's impact on the resultant stress in the native mitral valve [12] and post-implant hemodynamics through fluid–solid interaction analysis [13]. The optimal implantation position can be evaluated using computations based on echocardiography [14]. Additionally, machine learning, relying on simulations, is capable of predicting patient outcomes [15].

This study aimed to quantify the structural and hemodynamic performance of the MitraClip using a realistic and high-fidelity human heart geometry. Finite element analysis evaluated the stress in the clipped mitral valve, while computational fluid dynamics assess hemodynamic disturbances induced by the deformed valve shape. We demonstrated that in silico analysis emerges as a valuable tool to identify new metrics for clinicians in risk stratification after MitraClip implantation.

## 2. Materials and Methods

### 2.1. Left Heart and Mitral Valve Models

A geometric model of the left heart was derived from the two-chamber Living Heart Model (LHM) developed by the SIMULIA Living Heart Project within the 3D Experience platform (3D Experience R2023, Dassault Systèmes, Providence, RI, USA) [16,17]. This model accurately depicts an adult male heart with normal physiological function, as shown in Figure 1A. It comprises the left ventricle, left atrium, and the mitral valve with the chordae apparatus, providing a realistic and high-fidelity representation of the human anatomy. The initial geometry of the heart was at 70% ventricular diastole whilst the mitral valve was at the zero-pressure configuration.

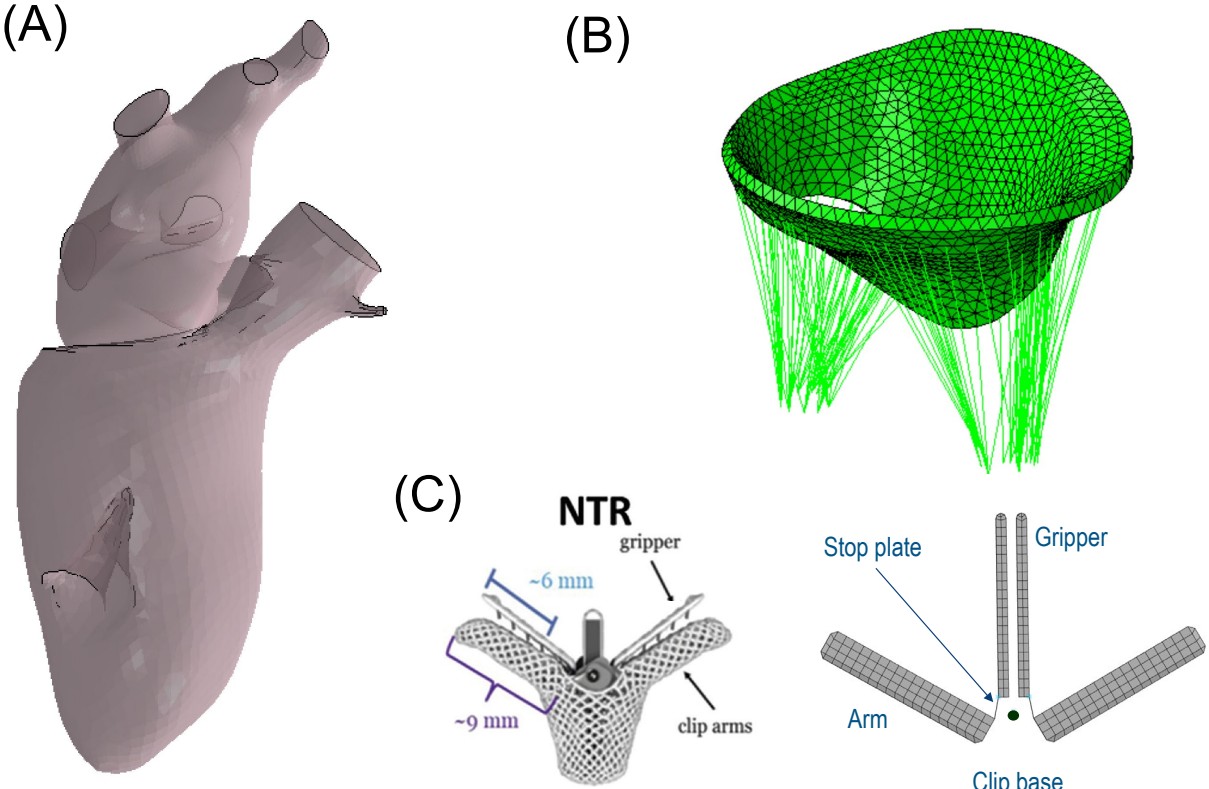

**Figure 1.** (**A**) geometric LHM of the left ventricle and left atrium; (**B**) mitral valve model showing the chordae apparatus; and (**C**) MitraClip and its model representation.

The biomechanical behavior of the left heart was considered with respect to both passive cardiac properties and active contraction originating from the myocardium. The passive material behavior was simulated using the anisotropic hyperelastic constitutive model developed by Ogden and Holzapfel [18], which has been widely used in many cardiac in silico analyses [17,19,20]. The deviatoric response was governed by the following strain energy potential:

$$\Psi_{dev} = \frac{a}{2b}exp[b(I_1 - 3)] + \sum_{i=f,s} \frac{a_i}{2b_i}\left\{exp\left[b_i(I_{4i} - 1)^2\right] - 1\right\} + \frac{a_{fs}}{2b_{fs}}\left[exp\left(b_{fs}I_{8fs}^2 - 1\right)\right] \quad (1)$$

Six material parameters, $a$, $b$, $a_i$, $b_i$, $a_{fs}$, and $b_{fs}$, and four strain invariants, $I_1$, $I_{4f}$, $I_{4s}$, and $I_{8fs}$, defined Equation (1) for the left heart (see Table 1). The initial values of $a_i$ and $b_i$ were determined from the calibration of myocardial tissue using experimental biaxial stretch and triaxial shear data [21,22]. This calibration was further refined by adjusting the initial values for the ventricle and atrium separately by scaling them to match the Klotz curve of the diastolic pressure–volume relationship [23]. The pressure of the Klotz curve was in the range from 0.1 mmHg to 4 mmHg. The active tissue response, capable of capturing the Frank–Starling effect, enabled the stress activation of the fiber and sheet directions in the constitutive model. This was accomplished by using a time-varying elastance model where the active force was a function of the current sarcomere length, peak intracellular calcium concentration, and fiber activation time, as described by Warrick et al. [24]. Ventricular myocardial fiber orientation was defined using a rule-based algorithm, with directions varying from $-60°$ on the epicardium to $+60°$ on the endocardium [25]. The heart model was spatially constrained by fixed node sets at the distal ends of the four pulmonary veins and the aortic root. A direct approach to computing the initial stress state was implemented for the left heart model, as described by Gee and collaborators [26]. With this approach, two simulations were performed. In the first simulation, a built-in VUMAT was used to compute the stretches of the pressure-free left heart configuration. The stretches from this simulation were then automatically extracted using a Python script and read in a second simulation that used constitutive laws to produce stresses in equilibrium with stretches. Fiber orientation in the atrium was established using an atlas-based method, which provided comprehensive maps of fibrous origins in the atria [27], indicating consistent fiber orientation across subjects. The atrium was divided into twelve regions, with each region being assigned a qualitatively consistent fiber orientation based on the atlas. No variation in fiber direction was defined in the through-thickness direction.

**Table 1.** Material parameters of mitral valve and left ventricle.

| | a (MPa) | b | a$_f$ (MPa) | b$_f$ | a$_s$ (MPa) | b$_s$ | a$_{fs}$ (MPa) | b$_{fs}$ | C$_{a0}$ (μmol/L) |
|---|---|---|---|---|---|---|---|---|---|
| Mitral Valve | 8.7e−4 | 2.7 | 5.0e−4 | 12.0 | 3.8e−3 | 7.6e−1 | 3.8e−3 | 7.6e−1 | |
| Left ventricle | 4.0e−1 | 12.0 | 5.0e−1 | 5.0 | 2.0e−1 | 2.0 | 1.1e−2 | 2.0 | 2.66 |

The mitral valve was modelled by a solid part with uniform thickness of 1.45 mm for both anterior and posterior valve leaflets. The anisotropic and hyperelastic constitutive behavior described by Equation (1) was adopted for the mitral valve, but not considering active myocardial contraction. Specifically, the parameters of the passive material response were based on the biaxial experiments performed on porcine mitral valves by May-Newman et al. [28]. Specifically, similar material parameters were applied to both the anterior and posterior mitral valve leaflets. These parameters were determined by averaging the data reported by May-Newman, who conducted distinct material characterizations on the two leaflets. A realistic chordae apparatus was developed using truss elements connected from the papillary muscles to the mitral valve leaflets (see Figure 1B). An Ogden material model for the passive behavior of the chordae was used with the parameter values reported by Zuo et al. [29]. The mitral valve was connected to the left heart model using tied-contact

conditions. All anatomic parts were meshed using 3D tetrahedral elements whilst the chordae tendineae were discretized with truss elements. Specifically, the mitral valve was meshed with 13,230 elements, the left atrium with 435,650 elements, and the left ventricle with 602,180 elements.

## 2.2. MitraClip Simulation

The geometric design of the clip device comprised four rigid plates, closely resembling the actual transcatheter MitraClip device developed by Abbott (Figure 1C). The dimensions were chosen to match the NTR version of the MitraClip, with the two arms having a width of 5 mm, length of 8 mm, and thickness of 1.5 mm. The gripper had a thickness of 0.7 mm. "Stop plates" were included in the model to limit the leaflet insertion length, similar to the actual clip device. The arms and grippers of the clip device were actuated to fully grasp the mitral valve leaflets along a single axis at a hinge point located at the base of the clip. To ensure the stiff behavior of the device, rigid material properties were assigned to both arms and grippers, and the material density was set to achieve a total mass of 0.67 g, in accordance with the manufacturer's design specifications. A reference node was generated to anchor the clip device in place near the mitral valve leaflets (black dots in Figure 1C). To simulate the MitraClip procedure, both rotations and translations were implemented with respect to the reference node, utilizing kinematic couplings between the parts in the ABAQUS/Explicit solver (v2022, Dassault Systèmes, USA). All degrees of freedom of the clip device were constrained to the reference point. The simulation began with the arms opened to their reference configuration (Figure 2A). Subsequently, the grippers were rotated by 52 degrees, causing them to face the device arms (Figure 2B) and thus capturing the mitral valve leaflets. Finally, the clip device was closed by simultaneously rigidly rotating both grippers and arms by −52 degrees relative to the reference point (Figure 2C). Such rotation values were derived from manufacturer guidelines and the consideration that the face of the device should be made as parallel as possible.

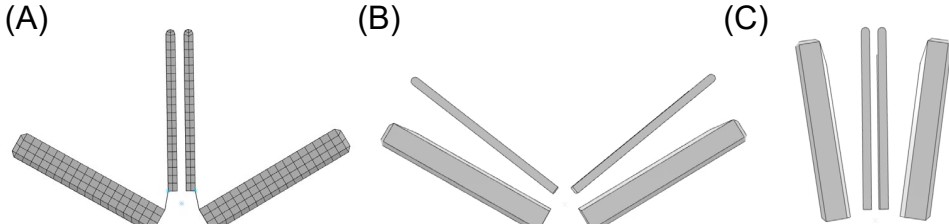

**Figure 2.** Simulation steps of clip device closure: (**A**) reference shape; (**B**) rotation of grippers for capturing the mitral valve leaflets; and (**C**) final clip closure to reduce mitral valve orifice area.

## 2.3. Post-MitraClip Flow Dynamic

The fluid dynamics of the left ventricle with the implanted clip device were assessed using the lattice Boltzmann method (LBM) in the XFlow solver (v2022, Dassault Systèmes, Providence, RI, USA). Unlike the traditional finite element method, LBM employs fictive particles to model fluid motion, undergoing propagation and collision processes over a discrete lattice grid. While the LBM approach is commonly used for engineering problems with transient flow [30], it has recently been proposed for biofluid mechanics applications [31,32]. The LBM automatically generates the grid in the solver before the mathematical solution, eliminating the need for a time-consuming meshing process during the preprocessing setup. This is particularly advantageous for complex geometries, such as the left heart with a clip device. The method is computationally efficient, even with turbulence modeling.

After simulating the MitraClip process, the deformed shapes of the left heart and clipped mitral valve were exported from ABAQUS as a stereolithographic file. To define the internal flow problem, the surface normal of each geometric part was oriented to point towards the interior domain of the left heart model. Blood was assumed to be a Newtonian and isothermal fluid, with a density of 1060 kg/m$^3$, viscosity of 0.00371 Pa·s, and a temper-

ature of 37 °C [33,34]. The turbulence model used was wall-adapting local eddy-viscosity (WALE). The flow simulation employed 749,220 unity D3Q27 lattices (74 × 74 × 135) with a high number of degrees of freedom per discrete element of fourth-order spatial discretization. A spatial length of 1 mm with a temporal resolution of 0.05 ms was chosen after a grid sensitivity study. Specifically, the peak velocity across the clipped mitral valve was monitored for four different grid refinements. No-slip boundaries were applied to the left ventricular and atrial walls. For the inlet, a flow velocity profile was set at each pulmonary vein and then split, assuming a proportionality of each pulmonary vein's cross-sectional area and mass balance conservation. The outflow had a pressure outlet profile imposed at the aortic root of the left ventricle (Figure 3). Two cardiac beats with a period of 1 s were simulated to reduce instabilities related to transient flow, and the last cycle was used for flow analysis. In order to compare hemodynamic patterns, fluid dynamics were also conducted on the mitral valve in its fully opened shape under healthy conditions.

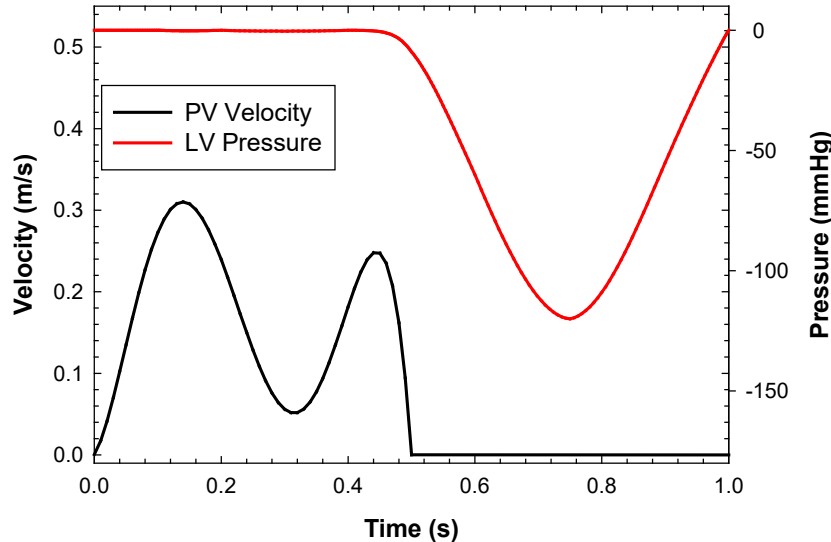

**Figure 3.** Inflow velocity and pressure outlet boundary conditions applied to the left heart model.

## 3. Results

Structural analysis using maximum principal stress revealed local maxima at the location of the implanted clip device in the mitral valve model (Figure 4). Upon the closure of the gripper component (Figure 4B), no significant changes in stress distribution on the mitral valve leaflets were observed. However, when the clip device was fully closed (Figure 4C), the mitral valve leaflets began to stretch, resulting in a notable increase in stress magnitude. The anterior mitral valve leaflet exhibited the highest stress distribution compared to the posterior valve leaflet.

Hemodynamics after the MitraClip procedure were investigated by analyzing flow streamlines over the cardiac cycle (Figure 5). A nested helical flow pattern was observed during the cardiac beating. At peak velocity, a high helical flow was observed at the left ventricular apex, with slower-moving fluid near the myocardial wall. During systole, the flow velocity decreased, and no significant backflow was observed from the clipped mitral valve to the left atrium.

A pressure drop of 3.5 mmHg was found across the clipped mitral valve at diastole, as shown in Figure 6. This pressure drop was likely induced by the reduced orifice area of the clipped mitral valve, leading to an increase in left atrial blood pressure.

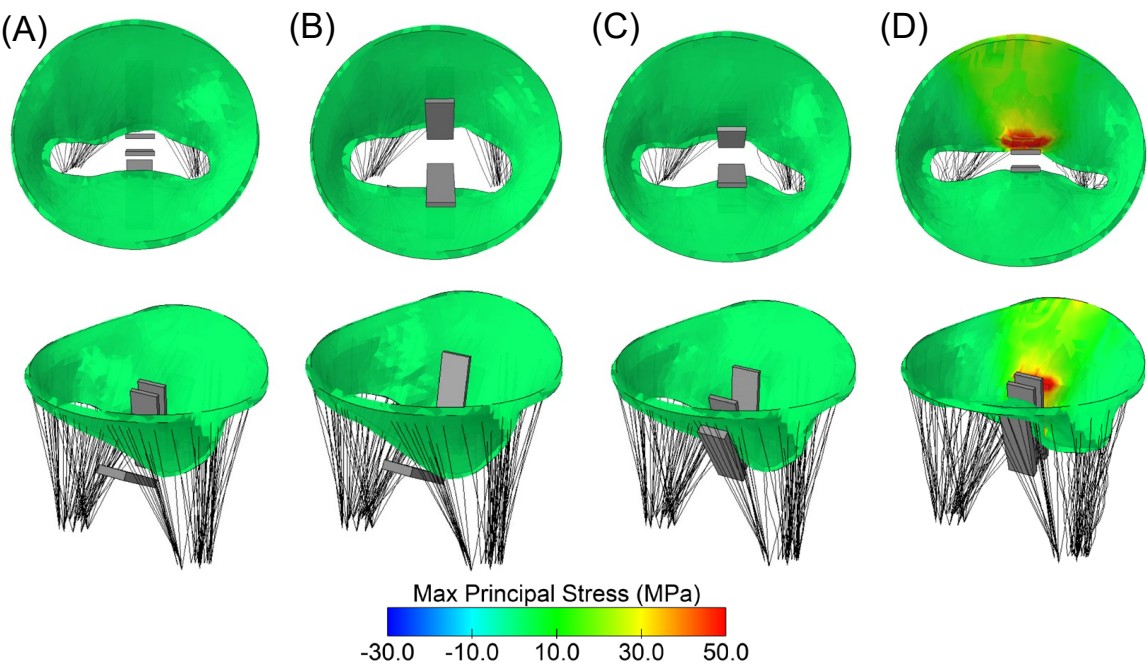

**Figure 4.** Different steps of clip device simulation: (**A**) reference configuration as loaded by uniform pressure distribution; (**B**) closure of clip grippers; (**C**) mid-closure of arms; and (**D**) full closure of the clip device.

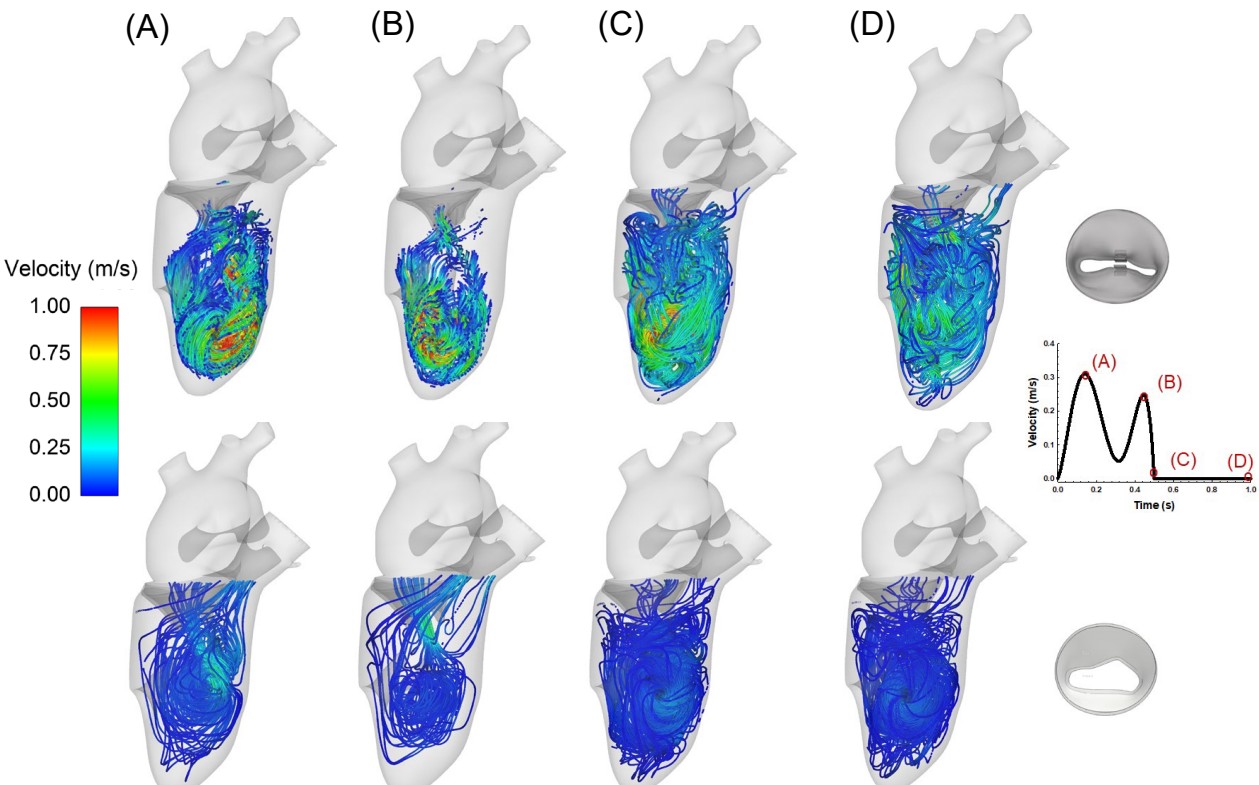

**Figure 5.** Flow velocity streamlines in the left ventricle after MitraClip (**top row**) and in healthy condition (**bottom row**); different phases of the cardiac cycle are highlighted from diastole to systole as shown by the inset on the bottom right.

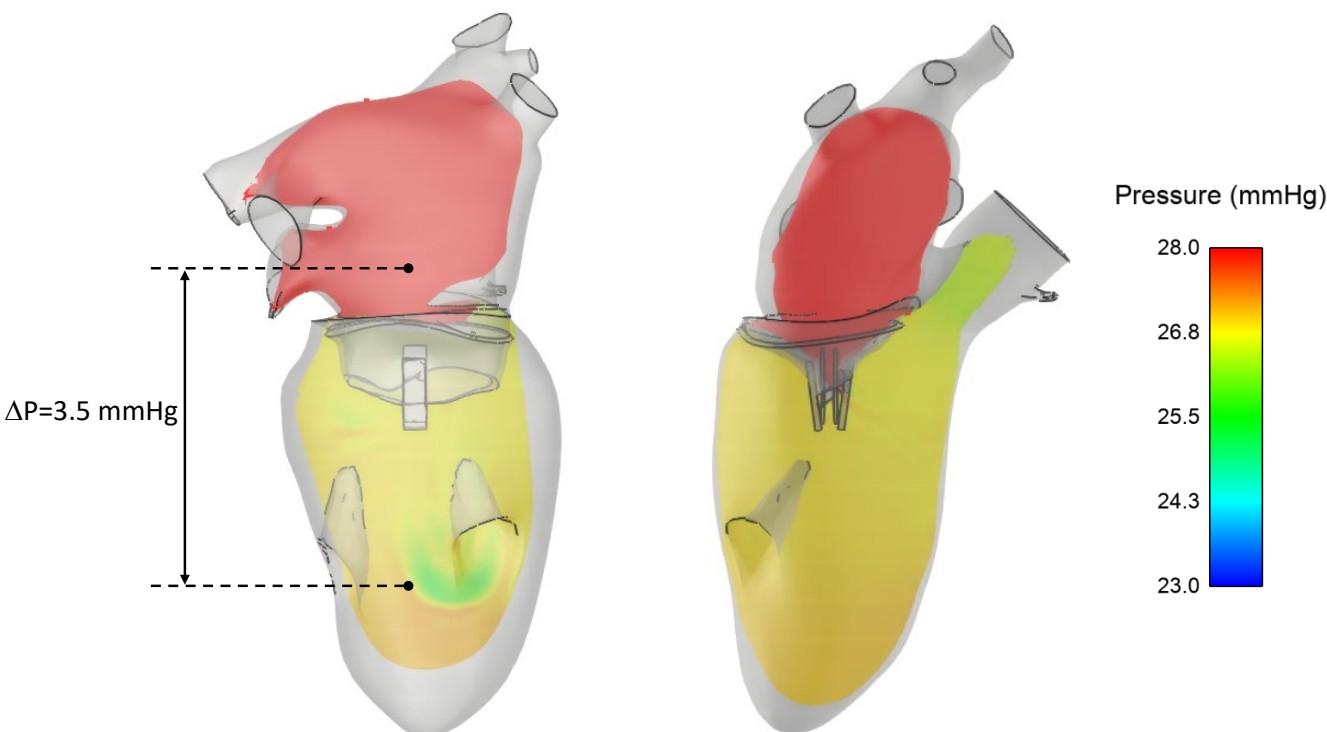

**Figure 6.** Blood pressure distribution in the left heart at diastole showing a pressure drop across the clipped mitral valve.

Analysis of the vortex in the fluid structure was performed using volumetric rendering of the vorticity parameter, defined as the modulus of the curl of fluid velocity (Figure 7). The residual mitral valve area led to the development of vortexes in the left ventricle, with pronounced flow curls primarily impinging the apical superior region of the left ventricle. The double orifice of the mitral valve exhibited a predominant vorticity structure on the left side of the MitraClip, indicating a potential hemodynamics impairment of the left ventricle.

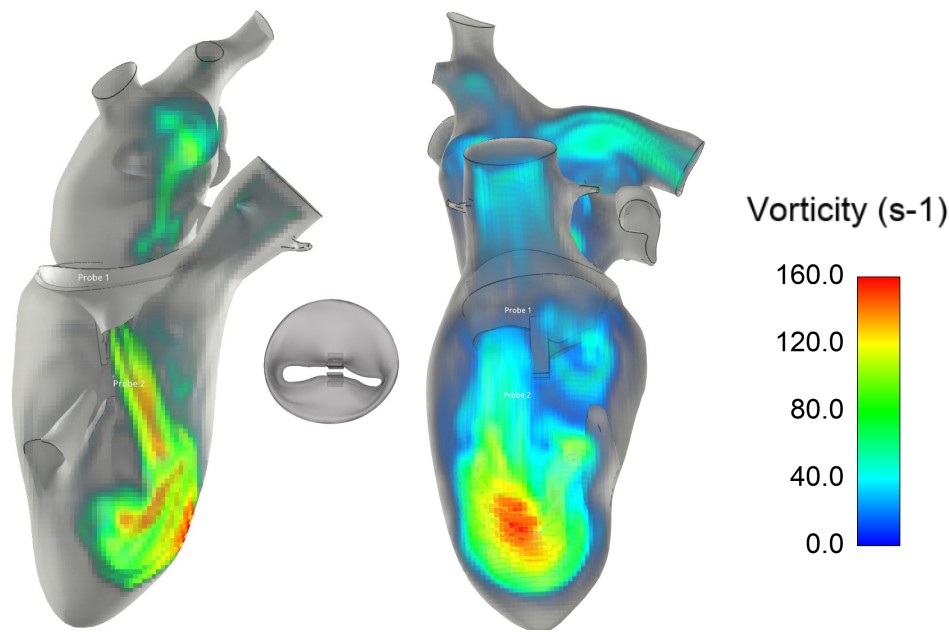

**Figure 7.** *Cont.*

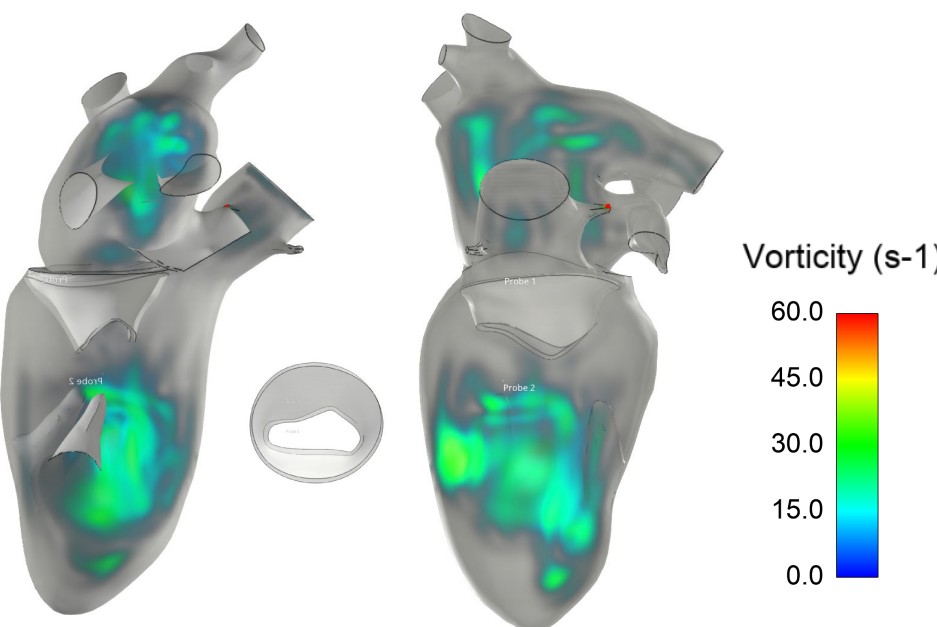

**Figure 7.** Vorticity structure inside the left ventricle after mitral clip (**top row**) and in healthy condition (**bottom row**).

## 4. Discussion

In this study, we utilized a realistic and high-fidelity model of the left heart to investigate the structural and hemodynamic performance of the MitraClip device. While the clip device effectively reduces regurgitant mitral flow, it also causes post-implant hemodynamic disturbances in the left ventricle due to the deformed state of the clipped valve leaflets. The main findings of our simulation, combining both finite element and computational flow analyses, can be summarized as follows:

1. MitraClip implantation leads to geometrical changes in the mitral valve, resulting in local maxima of principal stress in the regions of the valve leaflets that are constrained by the device.
2. Hemodynamic disturbances are observed in the left ventricle after MitraClip implantation and are characterized by nested helical flow and vorticity in the superior apical region.

These results underscore the significance of computational modeling in predicting potential complications and adverse events during the MitraClip procedure. By identifying these risks in advance, clinicians can develop strategies to mitigate them and improve patient outcomes. This study adds further evidence to the value of computational modeling as a crucial tool for evaluating and optimizing medical interventions for heart valve disorders.

There is a growing interest in applying computational tools to predict the interaction between devices designed for the mitral valve and the human host [35]. The use of computational modeling to assess the biomechanical implications of transcatheter mitral valve repairs not only enhances our understanding of post-implantation outcomes but also offers valuable insights for designing next-generation cardiovascular devices. In the case of edge-to-edge mitral valve repair, simulations of the surgical procedure were compared to echocardiography, revealing that the stress in the leaflets after the edge-to-edge suture was reduced [36]. In previous studies, we demonstrated the feasibility of transcatheter mitral valve replacement in the setting of failed annuloplasty band ring [37], surgical bioprosthesis [38], and massive mitral valve calcifications [39]. In computational studies of the MitraClip, Morgan and collaborators concluded that an uneven MitraClip position led to low anterior and posterior leaflet stress with respect to the model with leaflet prolapse and no clip [40]. This is in contrast to our findings and those of other researchers [41–43], which indicated increased stress in the clipped mitral valve. Using 4D transesophageal

echocardiography for mitral valve segmentation, Mansi et al. [41] developed a finite element model of the clip device and then compared the results to postoperative images. They found a good agreement between imaging and simulation results of MitraClip implantation. Sturla and collaborators [44] carried out both in vitro and in silico analyses of the MitraClip to fully exploit the versatility of mechanical stress analysis in a realistic scenario. They demonstrated that in silico analyses could pinpoint specific biomechanical implications and potentially support the pre-operative planning of the MitraClip in the diseased mitral valve. However, the majority of computational studies have primarily focused on the structural mechanics of MitraClip implantation. To the best of our knowledge, the only study reporting on hemodynamic alterations after MitraClip implantation is the one reported by Caballero et al. [45]. They adopted fluid–solid interaction analysis to demonstrate an increase in mitral valve leaflet stress and non-physiological flow in the left ventricle after MitraClip implantation. In our study, we corroborated their findings by employing a high-fidelity model of the left heart and the mitral valve with chordae tendineae. Specifically, we characterized the hemodynamics by observing a slow-moving nested helical flow near the left ventricular wall, alongside high flow velocities in the apex regions. Additionally, our analysis of the vortex structure further supported the value of computational flow analysis in revealing abnormal hemodynamic conditions induced by the double orifice area configuration of the mitral valve after MitraClip implantation.

These findings are crucial, as effective blood pressure management is vital for patients who have undergone the MitraClip procedure. Hypertension can strain the heart and worsen mitral regurgitation, potentially leading to the need to redo the MitraClip procedure or make a change in therapy. Continuous left atrium pressure monitoring may serve as a useful tool for procedural guidance during transcatheter mitral repair. Maor et al. [46] evinced that acute changes in the left atrium pressure after MitraClip implantation are associated with an improvement in exercise capacity. They also found an increase of 3 mmHg in blood pressure in the left atrium, consistent with the pressure drop found in the present computational study.

This study acknowledges its limitation in analyzing only one healthy mitral valve geometry. To account for the diseased valve state, modifications such as enlarging the annulus dimension or changing material properties would be necessary. The present findings are specific to the proposed geometric model, and no general conclusions can be drawn without considering variations in mitral valve anatomy and MitraClip implantation strategies. As a result, the computational framework employed in this study should in future be extended to include a larger patient cohort, to account for patient differences. The study method should also be extended to include different versions of the MitraClip device with dimensions differing from the current geometry. While the left heart model was constrained at the distal ends of the aortic root and pulmonary veins, the influence of the pericardium on heart kinematics was not taken into account. The proposed fiber angle of the left ventricle could likely have influenced cardiac motion in this study and needs further investigation. Additionally, the simulation results were not validated against imaging data of the MitraClip implanted in real patients. While this validation is crucial for ensuring the accuracy of the model, this study still offers valuable insights thanks to a systematic analysis of both structural and hemodynamic parameters resulting from MitraClip implantation in a realistic human model. Overall, this study provides a solid foundation for understanding the impact of MitraClip implantation on the human heart. However, further research is needed to incorporate variations in mitral valve pathology, validate the simulations against clinical data, and extend the analysis to a larger and more diverse patient population. Most importantly, future studies should consider the kinematics of the implanted MitraClip, enabling the utilization of fluid–solid interaction analysis for this specific purpose.

## 5. Conclusions

In this study, the structural and hemodynamic performance of the MitraClip device using a realistic and high-fidelity model of the human heart was investigated. The analysis focused on a healthy mitral valve geometry, limiting generalizations to diseased valve states. The findings revealed geometric changes in the mitral valve after MitraClip implantation, resulting in local stress maxima in the constrained leaflet regions. Hemodynamic disturbances, characterized by nested helical flow and vorticity, were observed in the left ventricle. Our study adds to the growing body of evidence on the importance of computational modeling in assessing the impact of MitraClip implantation on the cardiovascular system, shedding light on potential complications and aiding in the development of better treatment strategies for patients with mitral valve disorders.

**Funding:** The study was funded by the SiciliAn MicronanOTecH Research and Innovation CEnter SAMOTHRACE "(MUR, PNRR-M4C2, ECS_00000022), spoke 3—Università degli Studi di Palermo" S2-COMMs—Micro and Nanotechnologies for Smart and Sustainable Communities.

**Institutional Review Board Statement:** Not applicable.

**Informed Consent Statement:** Not applicable.

**Data Availability Statement:** The data presented in this study are available on request from the corresponding author.

**Conflicts of Interest:** The author declares no conflict of interest.

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
