# Peer review of "In Silico Analysis of the MitraClip in a Realistic Human Left Heart Model"

_prosthesis, doi:10.3390/prosthesis5030061_

Round 1
Reviewer 1 Report
1) Your literature review could benefit from including additional sources. There are several recent papers on this topic that have not been cited in your manuscript. It would also be valuable to consider the research conducted by recognized groups who have made significant contributions to this area over the years, as their publications provide important insights related to your study, and you didn't cite any of their papers.
2) The simulations you have conducted may benefit from a closer consideration of realism, particularly in terms of the size discrepancy between your clip and the mitral valve leaflets. In other words, your clip model is disproportionally too large (unrealistic) when compared to the leaflets. For this paper to be accepted you'd have to have to re-run your simulations with more realistic geometries.
Author Response
Reviewer #1
We thank the reviewer for his or her valuable comments. We have taken these comments into careful consideration when preparing the revised manuscript and feel that the critiques led directly to an improved submission. We hope that the reviewer agrees. All changes made to the document were highlighted in yellow.
1) Your literature review could benefit from including additional sources. There are several recent papers on this topic that have not been cited in your manuscript. It would also be valuable to consider the research conducted by recognized groups who have made significant contributions to this area over the years, as their publications provide important insights related to your study, and you didn't cite any of their papers.
Reply: We added several publication on the recent results on computational modeling of MitraClip. The following sentence was added in the Introduction:
“Simulations of the MitraClip demonstrate the device's impact on the resultant stress of the native mitral valve [9] and post-implant hemodynamics through fluid-solid interaction analysis [10]. The optimal implantation position can be evaluated using computations based on echocardiography [11]. Additionally, machine learning, relying on simulations, showcases its predictive capability for patient outcomes [12].”
- Kamakoti, R.; Dabiri, Y.; Wang, D.D.; Guccione, J.; Kassab, G.S. Numerical Simulations of MitraClip Placement: Clinical Implications. Scientific reports 2019, 9, 15823, doi:10.1038/s41598-019-52342-y.
- Zhang, Y.; Wang, V.Y.; Morgan, A.E.; Kim, J.; Handschumacher, M.D.; Moskowitz, C.S.; Levine, R.A.; Ge, L.; Guccione, J.M.; Weinsaft, J.W.; et al. Mechanical effects of MitraClip on leaflet stress and myocardial strain in functional mitral regurgitation - A finite element modeling study. PLoS One 2019, 14, e0223472, doi:10.1371/journal.pone.0223472.
- Hart, E.A.; De Bock, S.; De Beule, M.; Teske, A.J.; Chamuleau, S.A.J.; Voskuil, M.; Kraaijeveld, A.O. Transoesophageal echocardiography-based computational simulation of the mitral valve for MitraClip placement. EuroIntervention : journal of EuroPCR in collaboration with the Working Group on Interventional Cardiology of the European Society of Cardiology 2019, 15, e239-e241, doi:10.4244/EIJ-D-18-00644.
- Dabiri, Y.; Mahadevan, V.S.; Guccione, J.M.; Kassab, G.S. Machine learning used for simulation of MitraClip intervention: A proof-of-concept study. Front Genet 2023, 14, 1142446, doi:10.3389/fgene.2023.1142446.
2) The simulations you have conducted may benefit from a closer consideration of realism, particularly in terms of the size discrepancy between your clip and the mitral valve leaflets. In other words, your clip model is disproportionally too large (unrealistic) when compared to the leaflets. For this paper to be accepted you'd have to have to re-run your simulations with more realistic geometries.
Reply: The left heart and mitral valve model here used originates from a cardiac tools developed by Dassault Systems and is a realistic (1:1) model of the human heart. Additionally, the mitral clip dimensions is obtained from the manufacturer. The following sentence was added in the Study Limitation section to suggests that different mitral clip devices should be used to corroborate the present findings:
“The findings should also be extended to different versions of the MitraClip device with dimensions differing from the current geometry.”
Reviewer 2 Report
This study explores the mechanical and hemodynamic effects of the MitraClip intervention by combining a preexisting high-fidelity left-heart geometry with fluid dynamics. The figures are clear and support the findings of the study. There is indeed a clear need for approaches that couple solid and fluid analyses to predict possible adverse events post-surgery. However, in the opinion of the reviewer, this work is not acceptable for publication in its current form and requires major revisions.
General comments:
1. Despite the present study having the capability to simulate an untreated regurgitant case (a necessary control), there is no direct comparison to an unclipped MV. Inclusion of these results would provide a much stronger case for the mechanics and fluid patterns reported in the Results. Many limitations are acknowledged in the Discussion, but in the opinion of the reviewer, some of these, especially "analyzing only one healthy mitral valve" (Line 253), must be addressed and incorporated in order for it to have significant impact.
2. The use of the term "remodeling" implies a change in the microstructure of the leaflet tissues due to turnover of constituents and cellular processes, which are not considered in this study. To avoid confusion, the author is encouraged to use an alternative term referring simply to the altered deformation state of the MV.
Specific comments:
3. (Line 71) When defining the Holzapfel-Ogden constitutive model (Eq 1), the author states that "initial values" are fit to data from previous literature and iteratively adjusted to match the Klotz curve. However, these parameters are not reported, limiting the reproducibility of the study.
4. (Line 75) More detail is needed on how the full cardiac cycle was simulated and calibrated to determine the motion of the LV/LA, specifically accounting for the differences between filling during diastole and contraction during systole, despite referencing an active contraction constitutive model.
5. (Line 85) Use of the Holzapfel-Ogden model for the mitral valve leaflets is potentially inappropriate, since the leaflets have a multilayered composition distinct from the myocardium. Furthermore, the study by May-Newman et al. reports different behaviors for anterior and posterior leaflets, but it is not stated whether different properties were reproduced in the model. Again, no fitted parameters are reported in the manuscript. This may call the reported stress values in Figure 4 into question.
6. (Line 92) The author states meshing the geometry with 3D tet/truss elements, but does not report the mesh/element size. The meshes appearing in Figure 1B,C do not depict tetrahedra. Additionally, the heart wall is not shown at all in Figure 1A. It is ultimately unclear where the 3D tets are used.
7. (Line 114) How were the rotations of the MitraClip components determined?
8. (Line 133) The method of coupling the solid and fluid problems appears insufficient. Based on the text, the LHM anatomy with MitraClip engaged is exported and essentially "frozen" while solving the fluid dynamics for 2 cardiac cycles without accounting for any motion in the LV associated with a normal cardiac cycle. This would be inaccurate because of the large deformations of the filling/contracting LV and the coaptation of the MV leaflets. Consequently, the lack of LV and MV motion also calls the fluid results into question.
The author is encouraged to proofread this manuscript to improve the consistency, grammatical correctness, and clarity of English language. Some examples:
1. The device is incorrectly named "MitralClip" throughout the text and should be "MitraClip" in accordance with the cited literature.
2. Line 34: "mini-invasive" should be "minimally invasive".
3. Lines 61 & 98: "comprises" should be "is comprised of".
Author Response
Reviewer #2
We thank the reviewer for his or her valuable comments. We have taken these comments into careful consideration when preparing the revised manuscript and feel that the critiques led directly to an improved submission. We hope that the reviewer agrees. All changes made to the document were highlighted in yellow.
General comments:
- Despite the present study having the capability to simulate an untreated regurgitant case (a necessary control), there is no direct comparison to an unclipped MV. Inclusion of these results would provide a much stronger case for the mechanics and fluid patterns reported in the Results. Many limitations are acknowledged in the Discussion, but in the opinion of the reviewer, some of these, especially "analyzing only one healthy mitral valve" (Line 253), must be addressed and incorporated in order for it to have significant impact.
Reply: We thank the reviewer for his/her suggestion to add a control scenario. In this way, the Figure 5 and 7 show a comparison of the post-hemodynamic of the MitraClip with that of the physiological normal mitral valve. Specifically, we performed a flow analysis of the healthy mitral valve at fully opened shape as taken from the simulation of the cardiac beat of the healthy model. Please see the new Figure 5 and 7. The following text was added in the “Post-MitraClip Flow Dynamic”:
“To compare hemodynamic patterns, flow analysis was also performed on the mitral valve in its fully opened shape under healthy conditions.”
- The use of the term "remodeling" implies a change in the microstructure of the leaflet tissues due to turnover of constituents and cellular processes, which are not considered in this study. To avoid confusion, the author is encouraged to use an alternative term referring simply to the altered deformation state of the MV.
Reply: The term “remodeling” was deleted. We used the term “deformed” or “changes” as suggested by the reviewer.
Specific comments:
- (Line 71) When defining the Holzapfel-Ogden constitutive model (Eq 1), the author states that "initial values" are fit to data from previous literature and iteratively adjusted to match the Klotz curve. However, these parameters are not reported, limiting the reproducibility of the study.
Reply: A table with the material parameters was added in Section “Left Heart and Mitral Valve Model”.
- (Line 75) More detail is needed on how the full cardiac cycle was simulated and calibrated to determine the motion of the LV/LA, specifically accounting for the differences between filling during diastole and contraction during systole, despite referencing an active contraction constitutive model.
Reply: The following sentence was added to add more information on the pre-loading of the left hear model (see section “Left Heart and Mitral Valve Model”):
“A direct approach to compute the initial stress state has been implemented for the left heart model as described by Gee and collaborators [23]. With this approach two simulations are performed. In the first simulation a built-in VUMAT is used to compute stretches of the free-pressure left heart configuration. The stretches from this simulation are then automatically extracted using a python script and read in a second simulation that used the constitutive law to produce stresses in equilibrium with stretches.”
- (Line 85) Use of the Holzapfel-Ogden model for the mitral valve leaflets is potentially inappropriate, since the leaflets have a multilayered composition distinct from the myocardium. Furthermore, the study by May-Newman et al. reports different behaviors for anterior and posterior leaflets, but it is not stated whether different properties were reproduced in the model. Again, no fitted parameters are reported in the manuscript. This may call the reported stress values in Figure 4 into question.
Reply: Indeed, we used uniform material properties for both mitral valve leaflets. The following sentences was added in the “Left Heart and Mitral Valve Model”:
“Specifically, similar material parameters were applied to both the anterior and posterior mitral valve leaflets. These parameters were determined by averaging data reported by May-Newman, who conducted distinct material characterizations on the two leaflets”
- (Line 92) The author states meshing the geometry with 3D tet/truss elements, but does not report the mesh/element size. The meshes appearing in Figure 1B,C do not depict tetrahedra. Additionally, the heart wall is not shown at all in Figure 1A. It is ultimately unclear where the 3D tets are used.
Reply: Figure 1B was fixed to show the mesh of the mitral valve instead of CAD model. The mitral valve had 13230 elements while the left hear had 435650 elements. The following sentence was added in the “Left Heart and Mitral Valve Model” section:
“Specifically, the mitral valve was meshed with 13,230 elements, whereas the left atrium comprised 435,650 elements”.
- (Line 114) How were the rotations of the MitraClip components determined?
Reply: The rotation of MitraClip around the hinge point is determined based on manufacturer guidelines and the consideration that the face of the device should be made as parallel as possible. The following was added in the text:
“Such rotation values were derived from manufacturer guidelines and the consideration that the face of the device should be made as parallel as possible.”
- (Line 133) The method of coupling the solid and fluid problems appears insufficient. Based on the text, the LHM anatomy with MitraClip engaged is exported and essentially "frozen" while solving the fluid dynamics for 2 cardiac cycles without accounting for any motion in the LV associated with a normal cardiac cycle. This would be inaccurate because of the large deformations of the filling/contracting LV and the coaptation of the MV leaflets. Consequently, the lack of LV and MV motion also calls the fluid results into question.
Reply: We agree with the reviewer that the present fluid dynamic simulation did not consider the motion of the heart and device. Future studies will be performed to account this aspect using fluid-solid interaction analysis. The following sentence was added in the Study Limitation paragraph:
“Most importantly, future studies should consider the kinematics of the implanted MitraClip, enabling the utilization of fluid-solid interaction analysis for this specific purpose.”
Comments on the Quality of English Language
The author is encouraged to proofread this manuscript to improve the consistency, grammatical correctness, and clarity of English language. Some examples:
- The device is incorrectly named "MitralClip" throughout the text and should be "MitraClip" in accordance with the cited literature.
Reply: we thanks for the error and the device was renamed correctly.
- Line 34: "mini-invasive" should be "minimally invasive".
Reply: fixed.
- Lines 61 & 98: "comprises" should be "is comprised of".
Reply: fixed
Author Response
Reviewer #3
We thank the reviewer for his or her valuable comments. We have taken these comments into careful consideration when preparing the revised manuscript and feel that the critiques led directly to an improved submission. We hope that the reviewer agrees. All changes made to the document were highlighted in yellow.
The authors use a computational model to investigate how MitraClip, an intervention to resolve mitral valve regurgitation, affects blood flow and stresses of the mitral valve leaflets. The manuscript is well written and clear. However, there are some major concerns about the methods, model validation and the effect of some model choices on the validity of the results.
The methods are not detailed enough to guarantee reproducibility:
- Could the authors please specify what phase of the cardiac cycle the anatomy was
generated from?
Reply: the mitral valve anatomy was generated at free-stress configuration while the initial geometry for the heart represents 70% ventricular diastole. The following sentence was added in the “Left Heart and Mitral Valve Model” section.
“The initial geometry for the heart represents 70% ventricular diastole whilst the mitral valve is at the zero-pressure configuration. “
- What are the final values for the material parameters resulting from the calibration of the Holzapfel-Odgen law?
Reply: A table with the material parameters was added in the Section “Left Heart and Mitral Valve Model”.
- What is the pressure the authors used to compute the Klotz unloaded volume? Was the model unloaded? If so, with what method? Could the author please provide more details about the calibration process?
Reply: Simulation of the diastolic filling to match the Klozt curve with pressure in the range of 0.1-4 mmHg. The details to obtain the free-stress configuration were added in the “Left Heart and Mitral Valve Model” section:
“A direct approach to compute the initial stress state has been implemented for the left heart model as described by Gee and collaborators [23]. With this approach two simulations are performed. In the first simulation a built-in VUMAT is used to compute stretches of the free-pressure left heart configuration. The stretches from this simulation are then automatically extracted using a python script and read in a second simulation that used the constitutive law to produce stresses in equilibrium with stretches”
- Could the authors specify how they computed the fibres in the atria? At the moment, they specify only how they computed fibre orientation in the ventricles • To use the Holzapfel-Odgen law for the mitral valve as well, the authors should have defined a fibre orientation for the mitral valve as well. How was this defined?
Reply: In the “Left Heart and Mitral Valve Model”, The following sentences were added to describe how fibers in the atrium were computed:
“Fiber orientation in the atrium is established using an atlas-based method, which provides comprehensive maps of fibrous origins in the atria [24], indicating consistent fiber orientation across subjects. The atrium is divided into twelve regions, with each region being assigned qualitatively consistent fiber orientation based on the atlas. No variation in fiber direction is defined in the through-thickness direction.”
- The boundary conditions applied to the mechanics model constrain only the terminal nodes of the aortic root and of the pulmonary veins, discarding the effect of the pericardium. However, the pericardium plays an important role on the simulated motion of the model (e.g. physiological upwards and downwards motion of the base). Even in the absence of clinical data, the authors should consider showing and discussing the motion reproduced by the model, as incorrect motion will lead to incorrect mitral valve dynamics too
Reply: We agree with reviewers that the pericardium can affect the heart motion as well as that of the mitral valve. The following sentence was added in the Study Limitation section:
“While the left heart model was constrained at the distal ends of the aortic root and pulmonary veins, the influence of the pericardium on heart kinematics was not taken into account.”
The model results in the pre-MitraClip state should be validated and used as a control:
Reply: We thank the reviewer for his/her suggestion to add a control scenario. In this way, the Figure 5 and 7 show a comparison of the post-hemodynamic of the MitraClip with that of the physiological normal mitral valve. Specifically, we performed a flow analysis of the healthy mitral valve at fully opened shape as taken from the simulation of the cardiac beat of the healthy model. Please see the new Figure 5 and 7. The following text was added in the “Post-MitraClip Flow Dynamic”:
“To compare hemodynamic patterns, flow analysis was also performed on the mitral valve in its fully opened shape under healthy conditions.”
- The motion of the model should at least be shown to guarantee that the model is reproducing physiological motion of the base, as this will affect the mitral valve dynamics significantly • How was the stress distribution in the pre-intervention model? Was it uniform and comparable to literature clinical data/previous models?
Reply: The output of living heart model from Dassault Systems has been validated against clinical data (though not patient-specific). Specifically, the living heart model is able to reproduce realistic pressure volume loop, ventricular dimension, ventricular pressure. The motion of mitral valve in terms of orifice area and closing is comparable to clinical data. Sinche the model is not patient-specific, not any one-to-one comparison is feasible. Nothing was added if permitted by the reviewer.
- Figure 5: the flow should be compared to the pre-intervention state. Showing only the post-MitraClip results does not provide information about how the intervention changes the dynamics.
Reply: The flow patterns of the healthy mitral valve were added in Figure 5.
- Figure 6 and 7: the pressure drop and the vorticity should again be compared to the pre-MitraClip state. How different were these from the pre-intervention state?
Reply: The vorticity of the healthy mitral valve were added in Figure 7.
Minor concerns:
- What’s the prevalence of this disease in females vs males? Females have typically smaller hearts. How is this expected to change the results of the study? This should be added to the discussion when the authors talk about the limitations of the study concerning anatomical variability.
Reply: We did not find sex-based disparities in mitral regurgitation to justify that the present model is limited to adult male only. Nothing was added in the text, if permitted by the reviewer.
- Figure 1A: a clipped view of the geometry would be helpful to visualise the papillary muscles.
Reply: the shape of papillary muscles are shown well in all figures. Figure 1 was fixed according to other reviewer comments so that we prefer not to change this figure.
- 0.1 s for a cardiac period seems short. Maybe the authors meant 1 s?
Reply: the cardiac beat was 1 s and this was adjusted in the main text.
- The steepness of the fibre angle of the ventricles is likely to affect the motion reproduced by the model. The authors could investigate how the fibre orientation of the ventricles affect the results.
Reply: this is a good point. Future studies will be performed to quantify this aspect. Nothing was added in the text if permitted by the reviewer.
Round 2
Reviewer 1 Report
I guess I have to trust that the ratio between the heart and clip models is accurately represented, as you have clarified in the manuscript. Just out of curiosity as it's rare to publish as a single author, could you clarify the reason for excluding other co-authors, considering that you have PhD students in your team?
Furthermore (regarding my prior comment that "It would also be valuable to consider the research conducted by recognized groups who have made significant contributions to this area over the years"), I believe I have to be more specific: it would be beneficial to include references to recognized figures in this field, such as Drs. Sacks and Yoganathan (i.e., not me) who seem to be omitted from your current version of the document, e.g., 10.3390/biology9070173.
Author Response
Reviewer #1
We thank the reviewer for his or her valuable comments. We have taken these comments into careful consideration when preparing the revised manuscript and feel that the critiques led directly to an improved submission. We hope that the reviewer agrees. All changes made to the document were highlighted in yellow.
I guess I have to trust that the ratio between the heart and clip models is accurately represented, as you have clarified in the manuscript. Just out of curiosity as it's rare to publish as a single author, could you clarify the reason for excluding other co-authors, considering that you have PhD students in your team?
Reply: This work was initially started by a master student for the FE simulation of MitraClip. She has run the simulation (but not developed the model) with my supervision. The XFLow part was entirely developed by myself. For this reason, I did not prefer to include the master student.
Furthermore (regarding my prior comment that "It would also be valuable to consider the research conducted by recognized groups who have made significant contributions to this area over the years"), I believe I have to be more specific: it would be beneficial to include references to recognized figures in this field, such as Drs. Sacks and Yoganathan (i.e., not me) who seem to be omitted from your current version of the document, e.g., 10.3390/biology9070173.
Reply: The following three papers from Sacks and Yaganathan were added:
- Toma, M.; Einstein, D.R.; Kohli, K.; Caroll, S.L.; Bloodworth, C.H.t.; Cochran, R.P.; Kunzelman, K.S.; Yoganathan, A.P. Effect of Edge-to-Edge Mitral Valve Repair on Chordal Strain: Fluid-Structure Interaction Simulations. Biology (Basel) 2020, 9, doi:10.3390/biology9070173.
- Toma, M.; Singh-Gryzbon, S.; Frankini, E.; Wei, Z.A.; Yoganathan, A.P. Clinical Impact of Computational Heart Valve Models. Materials (Basel) 2022, 15, doi:10.3390/ma15093302.
- Sacks, M.S.; Enomoto, Y.; Graybill, J.R.; Merryman, W.D.; Zeeshan, A.; Yoganathan, A.P.; Levy, R.J.; Gorman, R.C.; Gorman, J.H., 3rd. In-vivo dynamic deformation of the mitral valve anterior leaflet. Ann Thorac Surg 2006, 82, 1369-1377, doi:10.1016/j.athoracsur.2006.03.117.
Reviewer 2 Report
The additional details added by the author address the initial concerns; I have some remaining questions/comments below:
Specific comments:
1. Comparisons to the unclipped MV seem irrelevant if the valve is open, since the purpose of the clip is primarily to prevent regurgitation when it is closed. Further justification/clarification for this comparison should be provided, if only for my own understanding.
2. A table of material parameters has been included with the caption relating to MV, LV, and "ischemic region." There is no mention of ischemic region in the manuscript otherwise, and ischemic myocardium is known to be mechanically different from normal tissue. The author must correct or clarify this.
Minor comments:
3. Mesh sizes for MV and LA are reported, but the LV is left out.
4. In Figure 5, the inset is not on the "bottom left," but on the right.
There are still instances of "mitralclip" or "mitral clip" in the manuscript. Additionally, the typo in the title will need to be corrected in the final stage.
There is an incorrect use of "comprised" when describing the meshes.
Author Response
Reviewer #2
We thank the reviewer for his or her valuable comments. We have taken these comments into careful consideration when preparing the revised manuscript and feel that the critiques led directly to an improved submission. We hope that the reviewer agrees. All changes made to the document were highlighted in yellow.
Specific comments:
- Comparisons to the unclipped MV seem irrelevant if the valve is open, since the purpose of the clip is primarily to prevent regurgitation when it is closed. Further justification/clarification for this comparison should be provided, if only for my own understanding.
Reply: I do understand that the comparison with an healthy mitral valve is not the proper one but at least the added model allows to capture the difference in the heamodynamic when the MitraClip is implanted. For instance, vorticity markedly increased with the mitralclip as compared to normal heart condition. It is clear that comparison should be performed with some clinical study but this is not feasible given the fact that the model is ideal.
- A table of material parameters has been included with the caption relating to MV, LV, and "ischemic region." There is no mention of ischemic region in the manuscript otherwise, and ischemic myocardium is known to be mechanically different from normal tissue. The author must correct or clarify this.
Reply: we thank the reviewer for this minor issues on the Table legend. The term “ischemic region” was deleted as we did not use this in our computational model.
Minor comments:
- Mesh sizes for MV and LA are reported, but the LV is left out.
Reply: fixed with the following sentence:
“Specifically, the mitral valve was meshed with 13,230 elements, the left atrium with 435,650 elements and the left ventricle with 602,180 elements.”
- In Figure 5, the inset is not on the "bottom left," but on the right.
Reply: fixed
Comments on the Quality of English Language
There are still instances of "mitralclip" or "mitral clip" in the manuscript. Additionally, the typo in the title will need to be corrected in the final stage.
Reply: fixed on the title and abstract.
There is an incorrect use of "comprised" when describing the meshes.
Reply: the term was deleted
Reviewer 3 Report
The authors addressed almost all comments from the previous review. Only a couple of minor additions are needed:
· The authors state in their reply: “Simulation of the diastolic filling to match the Klozt curve with pressure in the range of 0.1-4 mmHg”, but the pressure values were not added to the manuscript. Can the authors please add these to the manuscript?
· When the authors say “the living heart model is able to reproduce realistic pressure volume loop, ventricular dimension, ventricular pressure.”, they should provide values at least for the LV peak pressure and the LV ejection fraction. Although the values will not be patient-specific, the peak pressure should be about 120 mmHg and the EF of about 50%, consistent with healthy values. The values simulated by the model should be added, otherwise no validation at all is provided. If the validation was performed in another study, this should be specified.
· A sentence about the different fibres orientations should at least be added to the limitations, as this can be important effect on the results.
Author Response
Reviewer #3
We thank the reviewer for his or her valuable comments. We have taken these comments into careful consideration when preparing the revised manuscript and feel that the critiques led directly to an improved submission. We hope that the reviewer agrees. All changes made to the document were highlighted in yellow.
The authors state in their reply: “Simulation of the diastolic filling to match the Klozt curve with pressure in the range of 0.1-4 mmHg”, but the pressure values were not added to the manuscript. Can the authors please add these to the manuscript?
Reply: the following sentence was added in the text:
“The pressure for the Klotz curve was in the ranged from 0.1 mmHg to 4 mmHg”
- When the authors say “the living heart model is able to reproduce realistic pressure volume loop, ventricular dimension, ventricular pressure.”, they should provide values at least for the LV peak pressure and the LV ejection fraction. Although the values will not be patient-specific, the peak pressure should be about 120 mmHg and the EF of about 50%, consistent with healthy values. The values simulated by the model should be added, otherwise no validation at all is provided. If the validation was performed in another study, this should be specified.
Reply: Validation of volume loop, ventricular dimension and pressure for the healthy left heart of living heart project is reported in the following reference
- Baillargeon, B., et al. (2014). "The Living Heart Project: A robust and integrative simulator for human heart function." Eur J Mech A Solids 48: 38-47.
- A sentence about the different fibres orientations should at least be added to the limitations, as this can be important effect on the results.
Reply: The following was added in the Study limitation section:
“The proposed fiber angle of the left ventricle can likely influence the cardiac motion and needs further investigation.”